# Soluble Spike DNA Vaccine Provides Long-Term Protective Immunity against SARS-CoV-2 in Mice and Nonhuman Primates

**DOI:** 10.3390/vaccines9040307

**Published:** 2021-03-24

**Authors:** Yong Bok Seo, You Suk Suh, Ji In Ryu, Hwanhee Jang, Hanseul Oh, Bon-Sang Koo, Sang-Hwan Seo, Jung Joo Hong, Manki Song, Sung-Joo Kim, Young Chul Sung

**Affiliations:** 1Research Institute, SL VaxiGen Inc., Korea Bio Park, Seongnam 13488, Korea; ybseo@slvaxigen.com (Y.B.S.); jiryu@slvaxigen.com (J.I.R.); hhjang@slvaxigen.com (H.J.); 2Research Institute, Genexine Inc., Korea Bio Park, Seongnam 13488, Korea; yssuh@genexine.com; 3National Primate Research Centre, Korea Research Institute of Bioscience and Biotechnology, Cheongju, Chungcheongbuk 34141, Korea; seul3198@kribb.re.kr (H.O.); porco9@kribb.re.kr (B.-S.K.); hong75@kribb.re.kr (J.J.H.); 4Science Unit, International Vaccine Institute, Seoul 08826, Korea; SangHwan.Seo@ivi.int (S.-H.S.); mksong@ivi.int (M.S.); 5GenNBio Inc., Seoul 06026, Korea; sungjoo.kim@gennbio.com

**Keywords:** COVID-19, DNA vaccine

## Abstract

The unprecedented and rapid spread of SARS-CoV-2 (severe acute respiratory syndrome-coronavirus-2) has motivated the need for a rapidly producible and scalable vaccine. Here, we developed a synthetic soluble SARS-CoV-2 spike (S) DNA-based vaccine candidate, GX-19. In mice, immunization with GX-19 elicited not only S-specific systemic and pulmonary antibody responses but also Th1-biased T cell responses in a dose-dependent manner. GX-19-vaccinated nonhuman primates seroconverted rapidly and exhibited a detectable neutralizing antibody response as well as multifunctional CD4+ and CD8+ T cell responses. Notably, when the immunized nonhuman primates were challenged at 10 weeks after the last vaccination with GX-19, they had reduced viral loads in contrast to non-vaccinated primates as a control. These findings indicate that GX-19 vaccination provides a durable protective immune response and also support further development of GX-19 as a vaccine candidate for SARS-CoV-2.

## 1. Introduction

Severe acute respiratory syndrome coronavirus-2 (SARS-CoV-2) emerged towards the end of 2019 as a causative agent of pneumonia in the city of Wuhan in China [1]. Since its emergence, the global situation has been dynamically evolving, and on 30 January 2020, the World Health Organization declared coronavirus disease 2019 (COVID-19) as a public health emergency of international concern (PHEIC), and it was declared as a global pandemic on 11 March 2020. Disease symptoms range from mild flu-like to severe cases with life-threatening pneumonia [2]. Unlike its predecessors of novel betacoronavirus such as SARS-CoV and MERS-CoV, SARS-CoV-2 transmits efficiently from person to person. Due to the high transmissibility and extensive community spread, SARS-CoV-2 has already caused nearly 20 million infections and over 700,000 deaths as of August 2020 [3]. It is estimated that until ~60 to 70% of people develop herd immunity, achieving sufficient control of SARS-CoV-2 to resume normal activities is unlikely [4]. Therefore, the rapid development of vaccines or other immunomodulating agents [5] against SARS-CoV-2 is important to change the global epidemiology of this virus.

The four major structural proteins of SARS-CoV-2 are the spike (S) protein, envelope (E) protein, membrane (M) protein, and nucleocapsid (N) protein, which are essential for the virus assembly and infection [6]. The S protein is an attractive target for vaccine design because it plays crucial roles in receptor binding, fusion, and viral entry into the host cell during the infection process. Proteolytic cleavage of the S protein generates two subdomains, S1 and S2, that are responsible for host cell angiotensin-converting enzyme 2 (ACE2) receptor binding and host cell membrane fusion, respectively. While the S2 domain is conserved across human coronaviruses, the S1 domain is divergent across the coronaviruses [7]. It has recently been reported that a vaccine strategy based on the S antigen can prevent SARS-CoV-2 infection and disease in a mouse model [8]. Moreover, studies in rhesus macaques have shown that vaccine strategies based on the S antigen can prevent SARS-CoV-2 infection and disease in this relevant animal model [9,10], indicating that the S antigen may be sufficient as a vaccine immunogen to elicit SARS-CoV-2 protective immunity.

The urgent need for vaccines prompted an international response, with the development of more than 170 candidate SARS-CoV-2 vaccines within the first 7 months of 2020 [11]. In order to achieve an effective and rapid vaccine response against emerging viruses, a manufacturing and distribution platform that can shorten the time to product availability as well as rapid vaccine design is important. While it typically takes more than several months to produce cell lines and clinical-grade subunit proteins, nucleic acid vaccines can be produced in weeks [12,13]. In addition, DNA-based vaccines do not require a cold chain, making them a great alternative to the availability of important, life-saving vaccines in resource-poor areas of the world. There have been studies over the past decades to improve the efficacy of DNA vaccines, and as a result, many improvements in efficacy have been made [14]. In addition to prophylactic DNA vaccines against viral infections such as MERS-CoV and Zika virus, a therapeutic DNA vaccine against cancers has been demonstrated in several clinical trials [15,16,17,18,19,20,21].

Here, we explored the potential of a soluble SARS-CoV-2 S DNA vaccine candidate, designated GX-19. Vaccination with GX-19 elicits a concurrent humoral response and Th1-biased immune responses in both mice and nonhuman primate (NHP) models. Notably, the vaccine drives potent cellular and humoral responses in NHPs, including neutralizing antibodies that provide potent protective efficacy against SARS-CoV-2 infection. The data demonstrate that the immunogenicity of this DNA vaccine supports the clinical development to advance the development of this DNA vaccine in response to the current global health crisis.

## 2. Results

### 2.1. Construction and Immunogenicity of SARS-CoV-2 DNA Vaccines

The sequence for the SARS-CoV-2 S protein was generated after analysis of the S protein genomic sequence, which was retrieved from the NCBI SARS-CoV-2 resource. The full-length or entire ectodomain of the S gene was codon optimized for increased antigen expression in mammalian cells and the N-terminal tissue plasminogen activator (tPA) signal sequence was added. The insert was then subcloned into the pGX27 vector [22] and the resulting plasmid was designated as pGX27-S and pGX27-S_ΔTM_ (Figure 1a).

To assess the immunogenicity of two candidate DNA vaccines, we immunized six-week-old female BALB/c mice intramuscularly (IM) twice at 2-week intervals. As indicated in Figure 1b, immunization with both of the DNA vaccine candidates induced a robust S protein-specific antibody response compared to the control. Interestingly, there were higher antibody titers in the pGX27-S_ΔTM_ group than in the pGX27-S group at both post-prime and post-boost. These results are not consistent with the previous report that demonstrated a higher binding antibody in full-length S DNA-vaccinated macaques compared to S_ΔTM_ DNA-vaccinated macaques [9].

### 2.2. GX-19 Induces Strong Humoral and Cellular Immune Responses in Mice

pGX27-S_ΔTM_, named GX-19, was therefore selected as the vaccine candidate to advance to immunogenicity and efficacy studies. The immunization with GX-19 elicited significant serum IgG responses against the S protein in a dose-dependent manner (Figure 2a). GX-19 vaccination elicited an S-specific IgG response in the bronchoalveolar lavage (BAL) fluid (Figure 2b). Live virus neutralizing titers were also evaluated in BALB/C mice following the same GX-19 immunization regimen. Consistent with the S binding antibody response, neutralizing activity was elicited by GX-19 (Figure 2c).

Given that severe SARS-CoV is associated with a Th2-biased immune response with an inadequate Th1 response [23,24,25], we evaluated the balance of Th1 and Th2 responses. Although BALB/c mice tend to develop a more Th2-biased response after vaccination [26], GX-19 induced a Th1-biased immune response as indicated by higher IgG2a/b responses when compared to IgG1 regardless of vaccination doses (Figure 3a,b). We also directly measured cytokine patterns in vaccine-induced T cells by cytometric bead array. GX-19-induced T cells secreted large amounts of IFN-γ, TNF-α, and IL-2 but did not secrete IL-4 or IL-5 (Figure 3d). At 2 weeks after the boost vaccination, spleens were harvested and S-specific T cell responses were measured by IFN-γ ELISPOT. Mice immunized with 5, 15, and 45 μg of GX-19 exhibited a dose-dependent splenic T cell response with mean IFN-γ spots per 10^6^ cells of 542, 872, and 1932, respectively (Figure 3c). To gain further insight into the responses of GX-19-induced T cell responses, we also measured the frequency of T cells producing multiple cytokines by intracellular cytokine staining (ICS). GX-19 vaccination exhibited a significantly higher percentage of S-specific CD4^+^ T cells or CD8^+^ T cells secreting IFN-γ, TNF-α, and IL-2 (Figure 3e; Appendix A).

### 2.3. GX-19 Elicits Robust Humoral and Cellular Immune Responses in NHPs

To investigate whether GX-19 was capable of inducing immune responses in an NHP model, three macaques were vaccinated with electroporation (EP)-enhanced delivery at weeks 0, 3, and 5.5 (day 39) with GX-19. Blood was collected at weeks 0, 5.5, 8, and 13.5 (day 95) to monitor vaccine-induced immune responses. The binding ELISA results showed that all three macaques immunized with GX-19 seroconverted after a single immunization with anti-S IgG titers that tend to be increased by boosting vaccination (Figure 4a). In addition, sera collected at weeks 0, 5.5, 8, and 13.5 were further analyzed for neutralization of wild-type SARS-CoV-2 (Korea centers for disease control and prevention) by the 50% plaque reduction neutralization test (PRNT_50_). Three macaques immunized with GX-19 displayed elevated neutralization titers with mean PRNT_50_ titers of 1:285 (at week 5.5) and 1:996 (at week 8). However, the mean PRNT_50_ titers at week 13.5 (2 weeks before virus infection) were higher than the baseline level but were lower than those detected at week 8 (Figure 4b).

To determine the impact of GX-19 on cellular immune responses, ELISPOT analysis was used to measure T cell responses in the blood of the vaccinated macaques. Three macaques developed T cell responses after single immunization. In addition, all animals exhibited such elevated responses after boost vaccination, implicating that vaccine-induced cellular immune responses became progressively stronger in macaques during GX-19 vaccination (Figure 4c). To gain further insights into GX-19-induced T cell responses, we again measured the frequency of T cells secreting multiple cytokines by ICS. GX-19 vaccination exhibited a meaningful induction of S-specific CD4^+^ T cells and CD8^+^ T cells producing IFN-γ, TNF-α, and, to a lesser extent, IL-2 (Figure 4d; Appendix A).

### 2.4. GX-19 Provides Protective Benefits to NHPs from Wild-Type SARS-CoV-2 Infection

Induction of long-term immunological memory for T cell and B cell responses is important for effective vaccine development. Unlike other vaccine studies in which NHPs confirmed protective efficacy against SARS-CoV-2 infection at 4~6 weeks after the last vaccination [9,10,27,28,29], we evaluated the protective efficacy approximately 10 weeks after the last vaccination. Accordingly, macaques were challenged by multiple routes with a total dose of 2.6 × 10^7^ 50% tissue culture infectious doses (TCID_50_), at 10 weeks after the last vaccination. This challenge route and dose were determined based on a model development study in which we challenged macaques that had no previous exposure to the virus [30]. Two of the three naive macaques showed elevated body temperature after viral infection, while none of the GX-19-vaccinated animals exhibited such symptoms. Further, GX-I9-vaccinated macaques showed more rapid recovery from lymphocyte reduction than did unvaccinated counterparts (Appendix A). High levels of viral load were observed in the unvaccinated macaques (Figure 5a,b) with a median peak of 7.54 (range 6.66–8.01) log_10_ viral copies/mL in the nasal swab and a median peak of 6.18 (range 6.01–6.81) log_10_ viral copies/mL in the throat swab (Figure 5c,d). Contrarily, we detected lower levels of viral load in GX-19-vaccinated macaques, including 1.58 and 1.57 log_10_ reductions in the median peak viral load in the nasal swab and throat swab, respectively (Figure 5a–d). Since the peak viral load does not reflect the presence of total virus over time, the virus load was then calculated based on the area under the curve (AUC). GX-19-vaccinated macaques had a viral AUC of 6.02 ± 0.23 log_10_ in the nasal swab and 4.99 ± 0.45 log_10_ in the throat swab, respectively, showing 1.46 and 1.45 log_10_ decreases in the viral AUC compared to those from unvaccinated counterparts (Figure 5e,f). For the assessment of infectious viruses, TCID_50_ assay was performed for the nasopharyngeal swab and oropharyngeal swab samples on Vero cells. The infectious viral load showed similar patterns to the viral RNA load. Although the difference was not found to be statistically significant due to the insufficient number of animals, lower levels of infectious viral load were observed in GX-19-vaccinated macaques (Appendix A).

At 4 days post-virus inoculation, all animals were euthanized, and tissues were collected. Consistent with previous reports [30], SARS-CoV-2 infection caused moderate to severe inflammation, as evidenced by small airways and adjacent alveolar interstitia in non-vaccinated macaques. In vaccinated macaques, the viral challenge caused mild histopathologic changes compared to those in control macaques (Figure 5g). Further pathological examination of each lung lobe of the challenged macaques also revealed the improvement in the severity of interstitial pneumonia in GX-19-vaccinated macaques (Figure 5h).

## 3. Discussion

In this study, we demonstrated that GX-19 (pGX27-S_ΔTM_) exhibited a higher S-specific antibody response than pGX27-S. In addition, GX-19 could elicit SARS-CoV-2 S-specific Th1-biased T cell responses in mice and NHPs. Vaccination of GX-19 also conferred effective protection against SARS-CoV-2 challenge at 10 weeks following the last vaccination.

The low immunogenicity and protective efficacy of S_ΔTM_ were reported in a recent study on the evaluation of the protective efficacy of a DNA vaccine expressing various forms of the SARS-CoV-2 S protein [9]. Our results also show that, in contrast to the full-length S DNA, GX-19 not only induced high antibody responses in mice and NHPs but also showed effective protection against SARS-CoV-2 virus challenge in NHPs. In the previous study [9], the vaccines were administered without the EP method, the method that significantly enhanced the in vivo delivery efficacy of a DNA vaccine by 100–1000-fold [31], resulting in weak antibody responses or T cell responses. On the other hand, electroporation-enhanced GX-19 induced robust antibody and T cell responses. In addition, the extent of increase in the immune response depends on the efficiency of gene expression by the vector [32], and the pGX27 vector has about three times higher expression strength than the commercial vector (unpublished data). Here, we believe that the introduction of a high-expression vector [22] into GX-19 along with an effective EP delivery system resulted in the efficient protective effect against SARS-CoV-2 infection through the induction of strong immune responses. However, further studies will be needed to compare pGX27-S and pGX27-S_ΔTM_ in NHPs under our conditions to confirm this finding.

In this study, we observed that GX-19 induced concurrent antibody, CD4^+^ T cell, and CD8^+^ T cell responses in both mice and NHP models. Indeed, successful DNA vaccination effectively induces complete complementation of the immune responses, including humoral and cellular responses (CD8^+^ and Th1 cellular responses), similar to those achieved by live attenuated viruses [15,16,33,34]. This can be explained by the nature of the DNA vaccine, presumably because its mechanism of action involves both class I antigen-processing pathways (i.e., intracellular processing of viral proteins into peptides and subsequent loading onto MHC class I molecules) and class II antigen-processing pathways (i.e., specifically engineered in the S signal sequence that cause an increased export of viral surface antigens). Among T cell responses, the balanced Th1/Th2 responses are important because vaccine-associated enhanced respiratory disease (VAERD) is associated with Th2-biased immune responses. Indeed, immunopathologic complications characterized by Th2-biased immune responses have been reported in the animal models of the SARS-CoV and MERS-CoV challenge [23,35,36,37,38,39], and similar phenomena have been reported in clinics vaccinated with whole-inactivated virus vaccines against RSV and measles virus [40,41]. In addition, the importance of T cell responses has been highlighted by a recent study of asymptomatic and mild SARS-CoV-2 convalescence [42]. These results collectively suggest that vaccines capable of generating balanced antibody responses and T cell responses may be important in providing protection against SARS-CoV-2 diseases. Here, we show that GX-19 induces Th1-biased responses, suggesting DNA vaccination can avoid the Th2-biased immune response associated with VAERD. In fact, there were subtle pathologic changes in the SARS-CoV-2-infected GX-19 vaccine group. These results were also demonstrated in other respiratory infection DNA vaccines such as for SARS-CoV and MERS-CoV in [43], and MERS-CoV in [33]. This suggests that a DNA vaccine platform can be a good alternative in vaccine development for emerging infections where balanced T cell and antibody responses are important.

It is desirable that the SARS-CoV-2 vaccine can prevent infection or disease and induce long-term immunity. Virus-specific T cell responses play an important role in antiviral and disease control. Immune-modulatory cytokines (e.g., IFN-γ, TNF-α, and IL-2) released from virus-specific CD4^+^ T and CD8^+^ T cells play a key role in several antiviral responses and act in synergy with type I IFNs to inhibit viral replication [44,45]. Patients with impaired IFN-γ activity were reported to have 5-fold increased susceptibility to SARS [46]. Clinical cases of asymptomatic virus infection indicate that virus-specific T cells can control disease even in the absence of neutralizing antibodies [42,47]. Here, we show that GX-19 induces potent antigen-specific CD4^+^ and CD8^+^ T cell activation and robust release of immune-modulatory cytokines in mice and NHPs, indicating that GX-19 can effectively control the disease of SARS-CoV-2. Therefore, clinical cases in which antibody responses rapidly decrease and disappear after SARS-CoV-2 infection indicate the importance of vaccines that can induce long-term immunological memory [48,49,50]. GX-19 induced potent CD4^+^ and CD8^+^ T cells in both animal models and it may confer long-lasting immunity against coronaviruses as indicated in SARS survivors, where CD8^+^ T cell immunity persisted up to 11 years [44,51]. We observed the protective benefits against viral infection about 10 weeks after the last vaccination, along with an elevated antibody response 8 weeks after the last vaccination. This indicates that the GX-19-induced immune response is long-lasting and also provides protective benefits against viral infection. Lastly, in the previous reports of NHP infection of other vaccines, the viral loads were almost completely controlled by vaccination at the beginning of the infection [29,52] and mild histopathological changes were observed [27,52]. Similar to the previous reports, our results also show mild histopathological changes compared to the control NHPs. Unlike previous reports, however, our results do not provide complete virus control at the beginning of the infection. This difference can be explained by the difference in the viral challenge doses and the difference in the time of viral challenge after the last vaccination. NHPs were challenged with higher viral doses (2.6 × 10^7^ TCID_50_) than previous reports (2 × 10^4^~4 × 10^5^ TCID_50_) and had a longer period (10 weeks) from last vaccination to viral infection compared to previous reports (4~6 weeks). However, these conclusions are drawn from a limited number of animals, especially in NHP models, so there is a limit to making clear conclusions. Additional studies involving a greater number of subjects will be needed to assess the safety and durability of the immune response and to assess protection against viral infection.

One drawback of this study was that it did not show statistical significance between naïve and GX-19-vaccinated NHPs due to the small number of NHPs. In fact, as a result of post hoc analysis on the results of NHP viral infection, the post hoc power values of viral loads of nasal swabs and throat swabs were found to be about 0.40~0.56. Based on the results, we calculated the number of animals per group that can guarantee more than 80% power, and as a result, it was expected that statistical significance could be secured if the experiments were conducted with at least five subjects per group.

## 4. Methods

**DNA vaccine construction.** The SARS-CoV-2 DNA construct encodes the SARS-CoV-2 spike (S) protein sequence of the current reported sequence (GenBank: QHD43416.1) as in other research cases [9,29]. The vaccine insert was codon optimized and commercially synthesized (Thermo Fisher Scientific, Carlsbad, CA, USA). Synthetic genes were subcloned into high-expression vector pGX27 [22].

**Mouse immunizations.** Female BALB/c mice aged 6–8 weeks (Koatech, Pyeongtaek-sk, Gyeonggi-do, Republic of Korea) were immunized with GX-19 vaccine or pGX27 in a total volume of 50 µL of PBS into the tibialis anterior muscle with in vivo electroporation with OrbiJector^®^ (SL VAXiGEN Inc, Seongnam-si, Gyeonggi-do, Republic of Korea) at weeks 0 and 2. On days 14 and 28, blood or BAL (bronchoalveolar lavage) fluid was collected, and mice were sacrificed 14 days after final immunization.

**NHP immunizations and challenge.** Cynomolgus macaques that weighed between 3.4 and 4.4 kg and had a mean age of 5.6 years (age range, 4.1–7.6 years) were immunized with 3 mg GX-19 vaccine at week 0, week 3, and week 5.5. GX-19 vaccine was given intramuscularly, followed by electroporation using OrbiJector^®^ (SL VAXiGEN Inc.). Blood was collected immediately before the first immunization (week 0) and every 1–3 weeks thereafter through week 8, and sera and PBMCs (peripheral blood mononuclear cells) were isolated to evaluate the humoral or cellular immune responses, respectively. At 10 weeks after the last vaccination, immunized macaques were moved to the animal biosecurity level 3 (ABL-3) laboratory in the Korea National Primate Research Centre (KNPRC) at the Korea Research Institute of Bioscience and Biotechnology (KRIBB). All animals were challenged with total 2.6 × 10^7^ 50% tissue culture infectious doses/mL [TCID_50_]/mL SARS-CoV-2 virus, obtained from the National Culture Collection for Pathogens (accession number 43326), via combined routes (intratracheal (4 mL), oral (5 mL), conjunctival (0.5 mL), intranasal (1 mL), and intravenous route (2 mL)) as previously described [30]. After viral challenge, macaques were anesthetized with ketamine sodium (10 mg/kg) and tiletamine/zolazepam (5 mg/kg) for temperature measurement and sample collection. Blood, nasopharyngeal swabs, and oropharyngeal swabs were collected at 0, 1, 2, 3, and 4 days post-infection (dpi). Swab samples were centrifuged at 1600× *g* for 10 min and filtered with 0.2 μm pore size syringe filters for further virus quantification. Immunization procedures were approved by the Institutional Animal Care and Use Committee (IACUC permit number ORIENT-IACUC-20044), and challenging procedures were approved by KRIBB IACUC (permit number KRIBB-AEC-20178).

**Antigen binding ELISA.** Serum and BAL fluid collected at each time point were evaluated for binding titers. Ninety-six-well immunosorbent plates (NUNC) were coated with 1 μg/mL recombinant SARS-CoV-2 S1+S2 ECD protein (Sino Biological 40589-V08B1) and S1 protein (Sino Biological 40591-V08H) in PBS (phosphate-buffered saline) overnight at 4 °C. Plates were washed 3 times with 0.05% PBST (Tween 20 in PBS) and blocked with 5% skim milk in 0.05% PBST (SM) for 2–3 h at room temperature. Sera or BAL fluid were serially diluted in 5% SM, added to the wells, and incubated for 2 h at 37 °C. Following incubation, plates were washed 5 times with 0.05% PBST and then incubated with horseradish peroxidase (HRP)-conjugated anti-mouse IgG (Jackson ImmunoResearch Laboratories 115-035-003, West Grove, PA, USA), IgG1 (Jackson ImmunoResearch Laboratories 115-035-205, West Grove, PA, USA), IgG2a (Jackson ImmunoResearch Laboratories 115-035-206, West Grove, PA, USA), or IgG2b (Jackson ImmunoResearch Laboratories 115-035-207, West Grove, PA, USA) for the mouse sera/BAL or anti-monkey IgG (Bethyl Laborabories A140-102P, Montgomery, AL, USA) for the NHP sera for 1 h at 37 °C. After final wash, plates were developed using TMB solution (Surmodics TMBW-0100-01, Eden Prairie, MN, USA) and the reaction was stopped with 2N H_2_SO_4_. The plates were read at 450 nm by SpectraMax Plus384 (Molecular Devices, San Jose, CA, USA).

**Live virus neutralization assay.** Vero cells (1.5 × 10^4^ cells/well) were seeded on a 96-well plate (Nunc, 167008) and incubated at 37 °C and 5% CO_2_ for 16 h. Serum samples were inactivated via incubation at 56 °C for 30 min and mixed with SARS-CoV-2 (300 plaque forming unit/25 µL) and incubated at 37 °C and 5% CO_2_ for 30 min. The serum–virus mixture was treated on the Vero cells and incubated at 37 °C and 5% CO_2_ for 4 h. After incubation, the treated mixture was removed and the cells were washed using 100 µL of phosphate buffered saline (PBS) (Gibco, 10010-023, Carlbad, CA, USA). The Vero cells were fixed using 300 µL of 10% formalin solution (Sigma, F8775, St.Louis, MO, USA) by incubating at 4 °C for overnight. After washing, the Vero cells were permeabilized by adding 100 µL of ice-cold 100% methanol (Sigma, D7, St.Louis, MO, USA). After 10 min incubation at room-temperature, the methanol was removed and the Vero cells were washed using 100 µL PBS, blocked using 100 µL of blocking buffer (0.5% normal goat serum (Abcam, Ab7481, Cambridge, CB2 0AX, UK) + 0.1% Tween 20 (GenDEPOT, T9100-100, Houston, TX, USA) + 1% (w/v) Bovine serum albumin (Sigma, 803-100G, St.Louis, MO, USA) in PBS) and incubated at room temperature for 30 min incubation at room temperature. After removing the blocking buffer, 3000-fold diluted 100 µL of anti-SARS-CoV-2 NP rabbit mAb (Sino Biological, 40143-R001, Beijing, China) was added on the Vero cells and incubated at 37 °C for 1h. After removing the Ab solution on the Vero cells, the Vero cells were washed using 200 µL of PBS containing 0.1% Tween 20. 2000-fold diluted goat anti-rabbit IgG-HRP (Bio-Rad, 170-6515, Hercules, CA, USA) solution was treated on the Vero cells and incubated at 37 °C for 1h. After removing the goat anti-rabbit IgG-HRP solution, the Vero cells were washed using 200 µL of PNS containing 0.1% Tween 20. 30 µL of TrueBlue solution was added on the Vero cells and incubate at room temperature for 30 min. After removing the TrueBlue solution, the cells were air dried until completely dry. The numbers of focus of each well were read using CTL reader (Cellular Technology Ltd., Shaker Heights, OH) and the neutralizing Ab titers were calculated using Microsoft Excel and SoftMax (Version 5.4.1) (Molecular Devices, San Jose, CA, USA).

**IFN-γ ELISPOT.** For mouse samples, the Mouse IFN-γ ELISPOT set (BD 551083) was used as directed by the manufacturer. ELISPOT plates were coated with purified anti-mouse IFN-γ capture antibody and incubated overnight at 4 °C. Plates were washed and blocked for 2 h with RPMI + 10% FBS (R10 media, Hyclone, St.Louis, MO, USA). Five hundred thousand splenocytes were added to each well and stimulated for 24 h at 37 °C in 5% CO_2_ with R10 media (negative control), concanavalin A (positive control), or specific peptide antigens (2 μg/mL). Peptide pools consisted of 15-mer peptides overlapped by 11 amino acids and spanned the entire SARS-CoV-2 S protein (GenScript, Nanjing, China). After stimulation, the plates were washed and spots were developed according to the manufacturer’s instructions. For NHP samples, the Monkey IFN-γ ELISpot^PLUS^ kit (MABTECH 3421M-APT-10) was used as directed by the manufacturer. Two hundred thousand PBMCs were stimulated with peptide pools, and plates were washed and spots were developed according to the manufacturer’s instructions. Plates were scanned and counted on AID ELISPOT reader classic. Spot-forming unit (SFU) per million cells was calculated by subtracting the negative control wells.

**Intracellular cytokine staining.** For mouse samples, splenocytes were stimulated in R10 media with specific peptide pools or medium alone (DMSO control) for 12 h. After stimulation, cells were washed with PBS for subsequent immunostaining. Antibodies for staining cells were CD8 FITC (Biolegend 100706, San Diego, CA, USA), IL-2 PE (Biolegend 503808), CD4 PE-Cy7 (Biolegend 100528), IFN-γ APC (Biolegend 505810), TNF-α (Biolegend 506328), CD3 BV605 (Biolgend 100351), and Live/dead IR (Invitrogen L10119). For NHP samples, cryopreserved and thawed PBMCs were resuspended in R10 media and rested overnight at 37 °C, 5% CO_2_, and subsequently PBMCs were stimulated in R10 media with specific peptide pools or medium alone (DMSO control) in the presence of 1 μg/mL of α-CD28 (BD bioscience 555725) and α-CD49d (BD bioscience 555501, San Jose, CA, USA) for 12 h. After stimulation, cells were washed with PBS for subsequent immunostaining and polychromatic flowy cytometric analysis. Antibodies for staining cells were CD3 PE (BD bioscience 552127), CD4 PerCP-Cy5.5 (Biolegend 317428), CD8a PE-Cy7 (Biolegend 301012), IFN-γ APC (Biolegend 506510), TNF-α BV421 (Biolegend 502932), IL-2 BV605 (Biolegend 500332), and Live/dead Near-IR (Invitrogen L10119). Fluorescence-activated cell sorting analysis was accomplished by a Fortessa flow cytometer (BD bioscience), and the data were analyzed using FlowJo software. Background cytokine expression in the DMSO controls was subtracted from that measured in the S peptide pools.

**Cytokine profile analysis by cytometric bead array (CBA).** Five hundred thousand splenocytes were plated and stimulated in R10 media with peptide pools (15-mers with 11-mer overlaps) corresponding to the SARS-CoV-2 S proteins (2 μg/mL) or the medium only as negative control in 96-well plates. Culture supernatants were harvested 48 h after the stimulation and cytokines were quantitated by the BD^TM^ CBA Mouse Th1/Th2 Cytokine kit (BD Biosciences) according to manufacturer’s instructions.

**Virus identification and quantification.** Filtered swab samples were inoculated into Vero cells and incubated for 3 days at 37 °C, for virus isolation to calculate the values of TCID50/mL using the Reed and Muench method. The viral RNA genome was extracted from the supernatant using QIAamp Viral RNA Mini Kit (Qiagen, Germantown, MD, USA) and stored at −80 °C in the ABL-3 facility until use. RT-qPCR was performed with a primer set targeting partial regions of the ORF1b gene in the SARS-CoV-2 virus using the QIAGEN OneStep RT-PCR kit (Qiagen) as previously reported [53]. For all RT-qPCR analyses, SARS-CoV-2 RNA standard and negative samples were run in parallel for determination of virus copy number.

**Histological evaluation.** Six lobes of the lung samples (three lobes in the right and the left lung, namely upper, middle, and lower lobes) of infected macaques were fixed in 4% paraformaldehyde for a minimum of 7 days and embedded in paraffin, and 4- to 5-μm sections were stained with hematoxylin and eosin. Sections of lung were blindly examined microscopically and given a score for severity of interstitial pneumonia from 0 (normal) to 6 (severe diffuse interstitial pneumonia).

**Statistical analysis.** Analysis of virologic and immunologic data was performed using GraphPad Prism 5 (GraphPad Software, San Diego, CA, USA). Comparison of data between groups was performed using two-sided Mann–Whitney tests. *p*-Values < 0.05 were considered significant. Post hoc power analysis was performed on results of the NHPs viral infection, and viral loads of nasal swab and throat swab were used as evaluation variables. Post hoc power analysis was based on Mann–Whitney tests using PASS2020 (NCSS Statistical Software, Kaysville, UT, USA).

## 5. Conlusions

In summary, these results explain the promising immunogenicity of GX-19 and, in particular, support the clinical potential of GX-19 by showing the protective effect against infection from NHPs.

## Figures and Tables

**Figure 1 vaccines-09-00307-f001:**
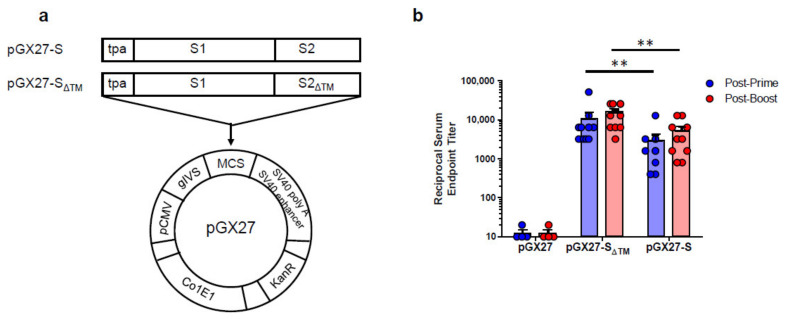
Diagram and immunogenicity of SARS-CoV-2 DNA vaccines. Schematic diagram of COVID-19 DNA vaccine expressing soluble SARS-CoV-2 S protein (S_ΔTM_) or full-length SARS-CoV-2 S protein (S) (**a**). BALB/c mice (*n* = 4–10/group) were immunized at weeks 0 and 2 with pGX27-S_ΔTM_, pGX27-S, or pGX27 (empty control vector) as described in the Methods. Sera were collected at 2 weeks post-prime (blue) and 2 weeks post-boost (red) and evaluated for SARS-CoV-2 S-specific IgG antibodies (**b**). All data are represented as individual values. ** *p* < 0.01 as determined by the Mann–Whitney test.

**Figure 2 vaccines-09-00307-f002:**
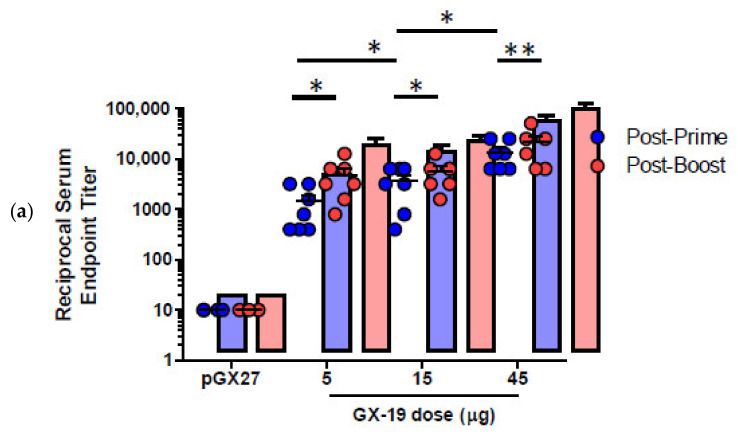
GX-19 elicits robust binding and neutralizing antibody responses in mice. BALB/c mice (*n* = 4–7/group) were immunized at weeks 0 and 2 with indicated doses of GX-19 or pGX27 as described in the Methods (**a**–**c**). Sera were collected at 2 weeks post-prime (blue) and 2 weeks post-boost (red) and assessed for SARS-CoV-2 S-specific IgG antibodies by ELISA (**a**), and for post-boost sera, neutralizing antibodies against SARS-CoV-2 live virus (**c**). Bronchoalveolar lavages (BALs) were collected at 2 weeks post-boost and assayed for SARS-CoV-2 S-specific IgG antibodies by ELISA (**b**). Data representative of two independent experiments. All data are represented as individual values. * *p* < 0.05, ** *p <* 0.01 as determined by the Mann–Whitney test.

**Figure 3 vaccines-09-00307-f003:**
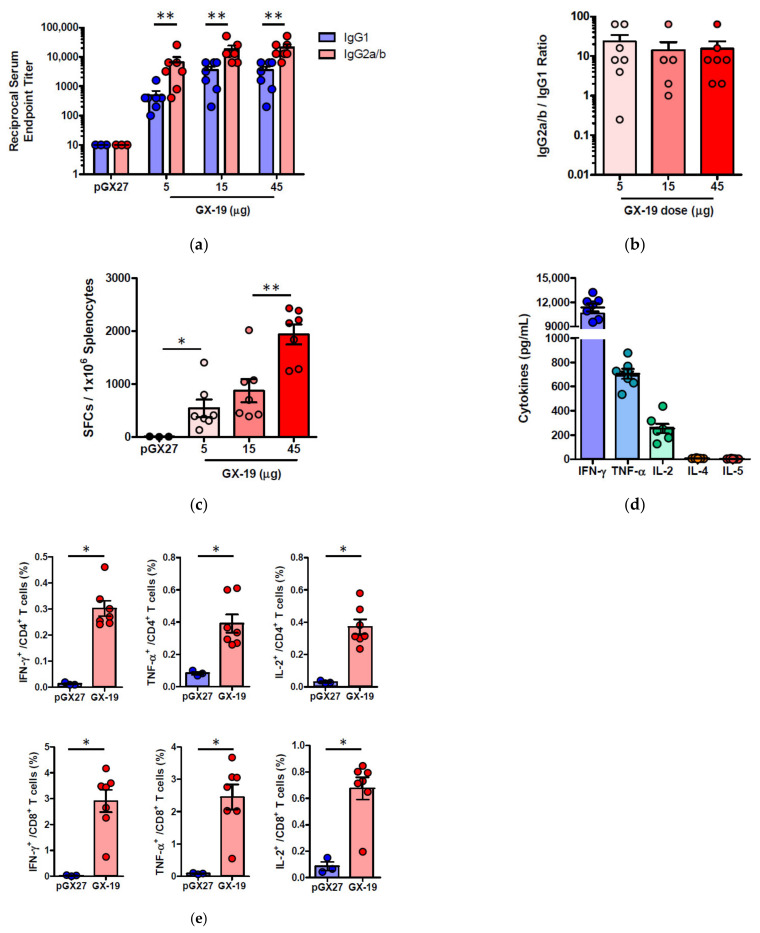
Immunization with GX-19 elicits Th1-biased T cell responses in mice. BALB/c mice (*n* = 3–7/group) were immunized at weeks 0 and 2 with indicated doses of GX-19 or pGX27 (empty control vector) as described in the Methods (**a**–**c**). Sera were collected at 2 weeks post-boost and assessed for SARS-CoV-2 S-specific IgG1 and IgG2a/b. Endpoint titers (**a**), and endpoint tier ratios of IgG2a/b to IgG1 (**b**) were calculated. At 2 weeks post-boost, mouse splenocytes were isolated and re-stimulated with peptide pools spanning the SARS-CoV-2 S protein ex vivo. Indicated cytokines in the supernatants of culture were quantified using a Th1/Th2 cytometric bead array kit. Mean value of the medium alone background (mean ± s.d., pg ml^−1^) was 19.17 ± 8.61 for IFN-*γ*, 57.12 ± 6.53 for TNF-α, 33.10 ± 6.72 for IL-2, 7.83 ± 0.45 for IL-4, and 4.66 ± 0.13 for IL-5 (**d**). T cell responses were measured by IFN-*γ* ELISPOT in splenocytes stimulated with peptide pools spanning the SARS-CoV-2 S protein. Shown are spot-forming cells (SFC) per 10^6^ splenocytes (**c**). Cells were stained for intracellular production of IFN-*γ*, TNF-α, and IL-2. Shown are the frequency of S-specific CD4^+^ or CD8^+^ T cells after subtraction of background (DMSO vehicle, Sigma-Aldrich, St. Louis, MO, USA) (**e**). Data representative of two independent experiments. All data are represented as individual values. * *p* < 0.05, ** *p* < 0.01 as determined by the Mann–Whitney test.

**Figure 4 vaccines-09-00307-f004:**
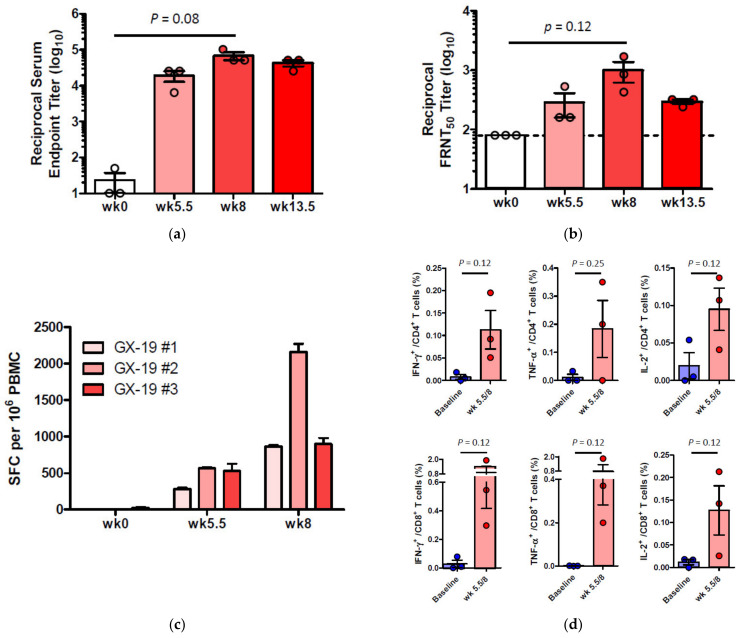
Antibody and T cell responses after GX-19 vaccination in macaques. Macaques (*n* = 3) were immunized with 3 mg of GX-19 as described in the Methods. Serum and PBMCs (peripheral blood mononuclear cells) were collected before (week 0), during (week 4 and 5.5), and after (week 8) vaccination and were assessed for SARS-CoV-2 S-specific IgG antibodies by ELISA (**a**) and neutralizing antibodies against SARS-CoV-2 live virus (**b**). Data represent mean SEM of individual macaques (GX-19 #1, GX-19 #2, GX-19 #3), and dashed line indicates the assay limits of detection. The number of SARS-CoV-2 S-specific IFN-*γ*-secreting cells in PBMCs was determined by IFN-*γ* ELISPOT assay after stimulation with peptide pools spanning the SARS-CoV-2 S protein. Shown are spot-forming cells (SFC) per 10^6^ PBMCS in triplicate wells (**c**). The frequency of S-specific CD4^+^ or CD8^+^ T cells producing IFN-*γ*, TNF-α, or IL-2 was determined by intracellular cytokine staining assays stimulated with SARS-CoV-2 S peptide pools. Shown are the frequency of S-specific CD4^+^ or CD8^+^ T cells after subtraction of background (DMSO vehicle) (**d**). Data of (**a**,**b**,**d**) are represented as individual values. *p*-Values determined by the Wilcoxon matched-pairs signed rank test; *p*-values are shown.

**Figure 5 vaccines-09-00307-f005:**
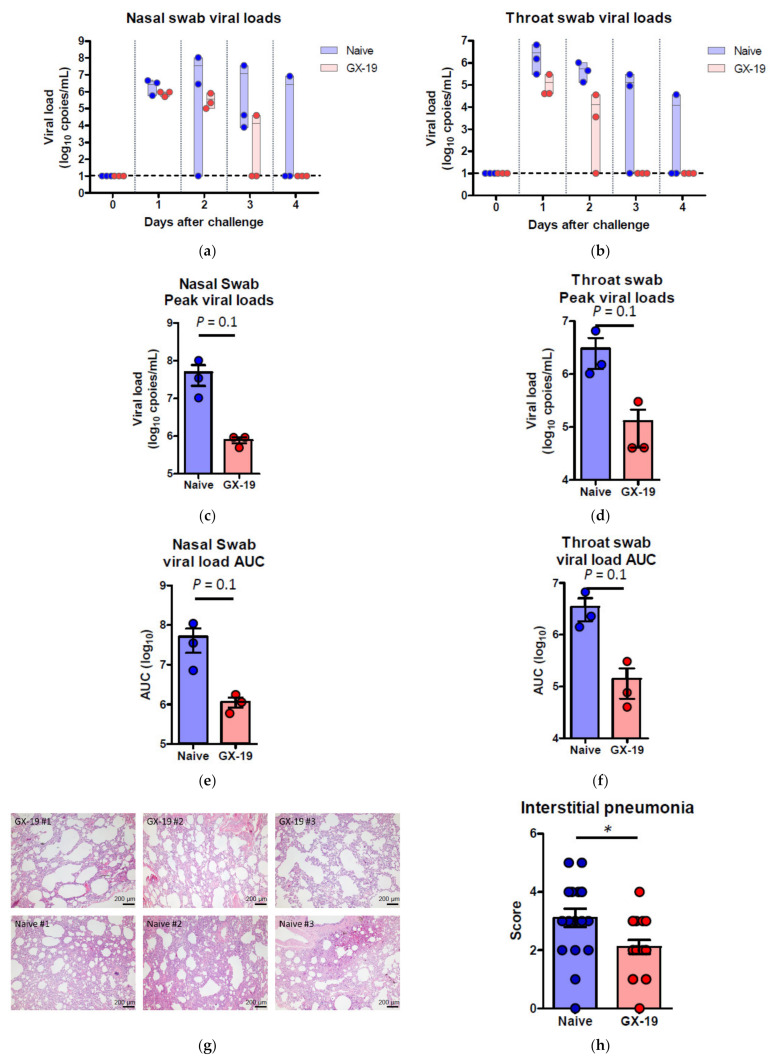
Protective efficacy of GX-19 against SARS-CoV-2 challenge. Non-vaccinated (*n* = 3, blue) and GX-19-vaccinated macaques (*n* = 3, red) were challenged by intratracheal, oral, conjunctival, intranasal, and intravenous administration of 2.7 × 10^7^ TCID_50_ SARS-CoV-2. Viral load was assessed in nasal swab (**a**) and throat swab (**b**) at multiple time points following challenge. Summary of peak viral loads and viral load area under the curve (AUC) in nasal swab (**c**,**e**) and throat swab (**d**,**f**) following challenge. Dashed line indicates the assay limit of detection. Histopathological changes in the lungs of SARS-CoV-2-challenged macaques (**g**). Interstitial pneumonia score by microscopic evaluation (*n* = 6 lung lobes of each animal per group) (**h**). The lung tissue sections were stained with hematoxylin and eosin (H&E). Data of (**a**–**f**) are represented as individual values. Data of (h) is represented as 6 lung lobes of each animal per group. *p*-Values determined by the Mann–Whitney test; *p*-values are shown * *p* < 0.05 as determined by the Mann–Whitney test.

## Data Availability

The data that support the findings of this study are available from the corresponding authors upon request.

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
