# Peer review of "Soluble Spike DNA Vaccine Provides Long-Term Protective Immunity against SARS-CoV-2 in Mice and Nonhuman Primates"

_vaccines, 2021, doi:10.3390/vaccines9040307_

Round 1

Reviewer 1 Report

The goal of this study was to determine the efficacy of a spike protein DNA vaccine in mice and macaques.  The results indicate that the DNA vaccine induced antibody and T cell responses, and in challenged macaques, prior vaccination reduced viral loads >10-fold with mild reduction in lung inflammation.  The work was conducted in a scientifically sound manner and the conclusions are supported by the results.  However, there are several problems.

  1. There is nothing new or surprising reported here compared to the previous report published by another group (Yu et al, Science 369:806-811, 2020). In that publication, a spike protein DNA vaccine was similarly tested in macaques and similar results were obtained.  In fact, they used a much larger number of macaques (35) compared to only 3 macaques used in the present manuscript and characterized the various effects on immunity and viral loads in much more detail.
  2. Given the wide availability of safe and effective RNA vaccines, it is less likely that a DNA vaccine will reach the market, especially since DNA vaccination against various pathogens has not been found to be protective in humans. The Yu et al, Science paper was published in August, 2020, at a time before the RNA vaccines were shown to be safe and effective.

Minor points:

  1. What is the pGX27-S truncated DNA construct?
  2. Panels B and C in Fig. 2 were not legible in the manuscript copy I received.

Author Response

Reviewer #1 (Reviewer’s comments are given in italic):

The goal of this study was to determine the efficacy of a spike protein DNA vaccine in mice and macaques.  The results indicate that the DNA vaccine induced antibody and T cell responses, and in challenged macaques, prior vaccination reduced viral loads >10-fold with mild reduction in lung inflammation.  The work was conducted in a scientifically sound manner and the conclusions are supported by the results.  However, there are several problems.

  1. There is nothing new or surprising reported here compared to the previous report published by another group (Yu et al, Science 369:806-811, 2020). In that publication, a spike protein DNA vaccine was similarly tested in macaques and similar results were obtained.  In fact, they used a much larger number of macaques (35) compared to only 3 macaques used in the present manuscript and characterized the various effects on immunity and viral loads in much more detail.

Response: As commented by the reviewer, it is similar to the ‘Yu et al, Science 369:806-811, 2020’ in evaluating the efficacy of SARS-CoV-2 DNA vaccine in animal models. However, unlike the results of the previous study (Yu et al, Science 369:806-811, 2020) that observed a weak antibody response in the S.dTM DNA (pGX27-SΔTM) vaccine, this study observed high antibody response in the S.dTM DNA (pGX27-SΔTM) vaccinated mice and NHPs. The possible explanations for this difference is described in the ‘Discussion section’ (Line 227-242).

  1. Given the wide availability of safe and effective RNA vaccines, it is less likely that a DNA vaccine will reach the market, especially since DNA vaccination against various pathogens has not been found to be protective in humans. The Yu et al, Science paper was published in August, 2020, at a time before the RNA vaccines were shown to be safe and effective.

Responses: Although RNA vaccines are safe and effective, but have limited in worldwide supply due to the very difficult transport and storage conditions. On the other hand, DNA vaccine have proven their safety in several clinical trials and have advantages in worldwide supply due to their convenient transport and storage properties. Although DNA vaccines are not currently approved for commercialization, their potential as a preventive or therapeutic vaccines has been demonstrated in recent years (DOI:https://doi.org/10.1016/S1473-3099(19)30266-X) (DOI:https://doi.org/10.1016/S0140-6736(17)33105-7) (DOI: 10.1056/NEJMoa1708120) (DOI:https://doi.org/10.1016/S0140-6736(15)00239-1) (DOI:https://doi.org/10.1016/S1470-2045(20)30486-1). Likewise, although the safety and efficacy of DNA vaccine against COVID-19 should be evaluated in several additional clinical trials, but the monkey results of previous (Yu et al, Science 369:806-811, 2020) and this study indicate that the possibility of success of DNA vaccine cannot be ruled out.

Minor points:

  1. What is the pGX27-S truncated DNA construct?

Response: ‘pGX27-S truncated DNA’ means that is was designed to express the ectodomain of SARS-CoV-2 spike antigen, and it is indicated as ‘pGX27-SΔTM (GX-19)’ in the manuscript (Line 76-80).

  1. Panels B and C in Fig. 2 were not legible in the manuscript copy I received.

Response: We have re-checked the figures.

Reviewer 2 Report

Bok Seo et al. studied the “Soluble spike DNA vaccine provides long-term protective immunity against SARS-CoV-2 in mice and nonhuman primates”.

In this study, the authors have attempted to develop novel synthetic SARS-CoV-2 spike DNA-based vaccine, i.e., GX-19. The authors have evaluated the vaccine candidate efficacy in two preclinical models, such as mice and nonhuman primates. Studies on mice have shown increased antigen-specific antibody titer, including Th1biased response. Studies on nonhuman primates have shown increased neutralizing antibodies with multifunctional T cell responses. Besides, GX-19 treatment also enhanced virus load clearance. Overall, the study is well organised, and the methods are clearly described. Statistics are appropriate.

Minor comments

Line 35-37: Please change the sentence “Therefore, the …….vaccines against SARS-CoV-2” to “Therefore, the …….vaccines or other immunomodulating agents (https://doi.org/10.1016/j.nantod.2020.101051) against SARS-CoV-2”

Figure 2b, 2c, 3d and 5b are not proper, please check.

Line 162: “spot-forming cells (SFC)” should first represent in figure 3 legend.

Line 263: Please change “VARED” to “VAERD”

Supplementary figure S1b: Please provide the “no peptide” conditions in the flow cytometry panel.

Supplementary figure S2b: Flow cytometry panels represent either “week 5.5 or 8”, please be precise.

Author Response

Reviewer #2 (Reviewer’s comments are given in italic):

  1. Line 35-37: Please change the sentence “Therefore, the …….vaccines against SARS-CoV-2” to “Therefore, the …….vaccines or other immunomodulating agents (https://doi.org/10.1016/j.nantod.2020.101051) against SARS-CoV-2”

Response: As suggested by the reviewer, we have change the sentence (Line 35-37)

  1. Figure 2b, 2c, 3d and 5b are not proper, please check.

Response: We have re-checked the figures.

  1. Line 162: “spot-forming cells (SFC)” should first represent in figure 3 legend.

Response: As suggested by the reviewer, we have re-written the sentence (Line 138)

  1. Line 263: Please change “VARED” to “VAERD”

Response: As suggested by the reviewer, we have changed the word (Line 262-263)

  1. Supplementary figure S1b: Please provide the “no peptide” conditions in the flow cytometry panel.

Response: As suggested by the reviewer, we have added ‘no peptide’ condition in the figure.

  1. Supplementary figure S2b: Flow cytometry panels represent either “week 5.5 or 8”, please be precise.

Response: As suggested by the reviewer, we have clarified time-point in the figure.

Round 2

Reviewer 1 Report

The revised manuscript inclues only very few changes and does not convincingly address my major concerns.

Author Response

Reviewer #1 (Reviewer’s comments are given in italic):

The goal of this study was to determine the efficacy of a spike protein DNA vaccine in mice and macaques.  The results indicate that the DNA vaccine induced antibody and T cell responses, and in challenged macaques, prior vaccination reduced viral loads >10-fold with mild reduction in lung inflammation.  The work was conducted in a scientifically sound manner and the conclusions are supported by the results.  However, there are several problems.

  1. There is nothing new or surprising reported here compared to the previous report published by another group (Yu et al, Science 369:806-811, 2020). In that publication, a spike protein DNA vaccine was similarly tested in macaques and similar results were obtained.  In fact, they used a much larger number of macaques (35) compared to only 3 macaques used in the present manuscript and characterized the various effects on immunity and viral loads in much more detail.

-> As commented by reviewer, it is similar to the previous study (Yu et al, Science 369:806-811, 2020) in that the efficacy of SARS-CoV-2 DNA vaccine was evaluated in an animal model. However, there are three differences from the previous study. First, this study evaluated pre-clinical efficacy of SARS-CoV-2 DNA vaccine candidate in both mice and NHPs. This can increase the likelihood of successful translational studies by enabling comparison of efficacy across species (small and large animal) for the same SARS-CoV-2 DNA vaccine candidate. Second, in this study, the NHPs was infected 10 weeks after the last vaccination, which is about 2 months longer than the previous study (3 weeks after the last immunization). We have shown that SARS-CoV-2 DNA vaccine-induced immune response is long-lasting, which differs from previous study (Line 178-184, Line 284-287). Lastly, the difference in antibody response induced by the soluble ectodomain (S.dTM) DNA vaccine construct. Unlike the results of the previous study that observed a weak antibody response in the S.dTM DNA (pGX27-SΔTM) vaccine, this study observed high antibody response in the S.dTM DNA (pGX27-SΔTM) vaccinated mice and NHPs. The possible explanations for this difference is described in the ‘Discussion section’ (Line 227-242). Additionally as commented by the reviewer, the limitation of this study due to the small number of animals was further described in the ‘Discussion section’ (Line 297-301).

  1. Given the wide availability of safe and effective RNA vaccines, it is less likely that a DNA vaccine will reach the market, especially since DNA vaccination against various pathogens has not been found to be protective in humans. The Yu et al, Science paper was published in August, 2020, at a time before the RNA vaccines were shown to be safe and effective.

-> The advantages of DNA vaccine and its potential as a preventive vaccines are described in the ‘Introduction section’ (Line 57-63). Although RNA vaccines are safe and effective, but have limited in worldwide supply due to the very difficult transport and storage conditions. On the other hand, DNA vaccine have proven their safety in several clinical trials and have advantages in worldwide supply due to their convenient transport and storage properties. Although DNA vaccines are not currently approved for commercialization, their potential as a preventive or therapeutic vaccines has been demonstrated in recent years. Likewise, although the safety and efficacy of DNA vaccine against COVID-19 should be evaluated in several additional clinical trials, but the monkey results of previous (Yu et al, Science 369:806-811, 2020) and this study indicate that the possibility of success of DNA vaccine cannot be ruled out.